# FedVARP: Tackling the Variance Due to Partial Client Participation in Federated Learning

**Divyansh Jhunjhunwala**[1]  **Pranay Sharma**[1]  **Aushim Nagarkatti**[1]  **Gauri Joshi**[1]

[1]Carnegie Mellon University, Pittsburgh, Pennsylvania, USA

## Abstract

Data-heterogeneous federated learning (FL) systems suffer from two significant sources of convergence error: 1) client drift error caused by performing multiple local optimization steps at clients, and 2) partial client participation error caused by the fact that only a small subset of the edge clients participate in every training round. We find that among these, only the former has received significant attention in the literature. To remedy this, we propose `FedVARP`, a novel variance reduction algorithm applied at the server that eliminates error due to partial client participation. To do so, the server simply maintains in memory the most recent update for each client and uses these as surrogate updates for the non-participating clients in every round. Further, to alleviate the memory requirement at the server, we propose a novel clustering-based variance reduction algorithm `ClusterFedVARP`. Unlike previously proposed methods, both `FedVARP` and `ClusterFedVARP` do not require additional computation at clients or communication of additional optimization parameters. Through extensive experiments, we show that `FedVARP` outperforms state-of-the-art methods, and `ClusterFedVARP` achieves performance comparable to `FedVARP` with much less memory requirements.

## 1 INTRODUCTION

Large-scale machine learning applications rely on numerous edge-devices to contribute their data, to learn better performing models. Federated Learning (FL) is a recent paradigm [Konečnỳ et al., 2016, McMahan et al., 2017] for distributed learning in which a *central server* offloads some of the computation to the edge-devices or *clients*, and the clients in return get to retain their private data, while only communicating the locally learned model to the server. For instance, when training a next-word prediction model [Hard et al., 2018], FL allows a client to enjoy suggestions supplied by thousands of other clients in the same federation without ever explicitly revealing its own personal text history.

Typical FL applications are targeted towards low-power mobile phones that have severely limited uplink (client to server) bandwidth. This necessitates the need for novel algorithms to reduce the *frequency* of communication required to train FL models. The first and the most popular algorithm in this setting is `FedAvg` [McMahan et al., 2017], which reduces communication frequency by requiring clients to perform *multiple* local computations in each round. In each round of `FedAvg`, clients first download the current global model, and run several steps of SGD on their private data before sending back their local updates to the server. The server then updates the global model using the average of the local updates sent by the clients.

A subtle yet important feature that distinguishes FL systems from traditional data-center settings is the presence of *heterogeneity* in local data across clients. While `FedAvg` improves communication-efficiency at the clients, it also leads to an additional error caused by this heterogeneity, colloquially known as *client drift* error [Karimireddy et al., 2019]. Informally, allowing clients to perform multiple local steps causes local models to drift towards their individual local minimizers, which is inconsistent with the server objective of minimizing the global empirical loss [Khaled et al., 2020, Wang and Joshi, 2021, Stich, 2019]. Despite recent advances [Pathak and Wainwright, 2020, Woodworth et al., 2020], a comprehensive theory regarding the usefulness of local steps remains elusive. Nonetheless, performing multiple local steps remains the most popular option for clients participating in FL due to its superior performance in practice.

Another defining characteristic of FL systems is *partial client participation*. Given the scale of FL [Kairouz et al.,

*Accepted for the 38th Conference on Uncertainty in Artificial Intelligence* (UAI 2022).

2019], it is unrealistic to expect *all* the clients to participate in every single round of FL training. For instance, clients may participate only when they are plugged into a power source and have access to a reliable wifi connection [McMahan et al., 2017]. In practice, we observe that only a small fraction of the total number of clients participate in any given round. This variance in client participation gives rise to what we term as *partial client participation error*. This error further compounds the effect of data heterogeneity as the global model is consistently skewed towards the data distributions of the participating clients in every round.

While error due to client drift has been well-established [Karimireddy et al., 2019, Acar et al., 2021, Khaled et al., 2020], we find that partial client participation error has not received similar attention. This is seen by the fact that several methods for mitigating client drift such as [Pathak and Wainwright, 2020, Zhang et al., 2020] cannot be directly extended to the partial client participation case. This is surprising, as our results indicate that error due to partial participation, rather than client drift, *dominates* the convergence rate of FedAvg (Theorem 1). For smooth nonconvex functions, we quantify the effect of the various noise sources (stochastic gradient noise, partial client participation, and data heterogeneity across clients) on the error floor of FedAvg, and observe that the dominant error is contributed by partial client participation.

**Our Contributions.** Keeping in mind the observation that partial client participation is the dominant source of error, we design a novel aggregation strategy at the server that completely eliminates partial client participation error. *Our algorithm keeps the local SGD procedure unchanged and only modifies the server aggregation strategy.* As a result, our approach does not introduce any extra computation at the clients or lead to any additional communication between the clients and the aggregating server. Furthermore, we also design a more server-friendly approach to our algorithm that allows the server to flexibly choose the amount of error reduction based on its system constraints. We summarize our main contributions below.

- We analyze the convergence of FedAvg and highlight that the dominant term in the asymptotic error floor comes from the partial participation of clients.
- In Section 3, we propose FedVARP (Federated VAriance Reduction for Partial Client participation), a novel aggregation strategy applied at the server to eliminate partial participation variance. FedVARP uses the fact that the server can store and reuse the *most recent update* for each client as an approximation of its current update. This allows the server to factor in contributions even from the non-participating clients when updating the global model.
- To relax the storage requirements of FedVARP, we devise a novel clustering based aggregation strategy called ClusterFedVARP in Section 4. ClusterFedVARP

in based on the observation that instead of storing unique latest updates for each client, we can cluster clients and store a single unified update that applies to *all* the clients in that cluster. We show that as long as the heterogeneity within a cluster is sufficiently bounded, ClusterFedVARP can significantly reduce partial client participation error, while being more storage-efficient.

- We conduct extensive experiments on vision and language modeling FL tasks that demonstrate the superior performance of FedVARP over existing state-of-the-art methods. Further, we show that ClusterFedVARP performs comparably to FedVARP, with much less storage requirements in practice.

For the purpose of theoretical analysis, throughout this paper we assume that in each round, the server uniformly selects a subset of clients from the total pool of clients. In practice, our algorithms can also be combined with non-uniform and biased client sampling strategies [Cho et al., 2020, Chen et al., 2020] for greater empirical benefits. Furthermore we note that the idea of reusing client updates has also been considered in a recent work MIFA [Gu et al., 2021], albeit in the context of dealing with arbitrary client participation. Owing to this similarity, we have a detailed comparison of our algorithm with MIFA in Section 3.1. While outside the scope of this work, we believe designing server aggregation strategies to deal with arbitrary client participation is an open and challenging direction for future work.

## 2 PROBLEM SETUP

We use the following notations in the remainder of the paper. Given a positive integer $m$, the set of numbers $\{1, 2, \ldots, m\}$ is denoted by $[m]$. Lowercase bold letters, for e.g., $\mathbf{x}, \mathbf{y}$, are used for vectors. Vectors at client $i$ are denoted with subscript $i$, for e.g., $\mathbf{x}_i$. Vectors at time $t$ are denoted with superscript $t$, for e.g., $\mathbf{y}^{(t)}$.

We consider optimizing the following finite sum of functions in a Federated Learning (FL) setting.

$$\min_{\mathbf{w} \in \mathbb{R}^d} f(\mathbf{w}) = \frac{1}{N} \sum_{i=1}^{N} f_i(\mathbf{w}) \tag{1}$$

where $f_i(\mathbf{w}) \triangleq \mathbb{E}_{\xi_i \sim \mathcal{D}_i} [\ell(\mathbf{w}, \xi_i)]$ is the local objective of the $i$-th client. Here $\ell(\cdot, \cdot)$ is the loss function, and $\xi_i$ represents a random data sample from the local data distribution $\mathcal{D}_i$. $N$ is the total number of clients in the FL system. Note that our formulation can be easily extended to the case where client objectives $\{f_i(\cdot)\}$ are unequally weighted.

We begin by recalling the FedAvg algorithm. At round $t$, the server selects a random subset of clients $\mathcal{S}^{(t)}$ and sends the global model $\mathbf{w}^{(t)}$ to these clients. The selected clients run LocalSGD (Algorithm 1) for $\tau$ steps. These clients

then send back their updates $\Delta_i^{(t)} = (\mathbf{w}^{(t)} - \mathbf{w}_i^{(t,\tau)})/\eta_c\tau$ to the server ($\eta_c$ is the client learning rate), which aggregates them to update the global model as follows:

$$\mathbf{w}^{(t+1)} = \mathbf{w}^{(t)} - \tilde{\eta}_s \frac{1}{|\mathcal{S}^{(t)}|} \sum_{i \in \mathcal{S}^{(t)}} \Delta_i^{(t)} \qquad (2)$$

where $\tilde{\eta}_s = \eta_s\eta_c\tau$, with $\eta_s$ being the server learning rate.

---

**Algorithm 1** `LocalSGD`$(i, \mathbf{w}^{(t)}, \tau, \eta_c)$

---

1: Set $\mathbf{w}_i^{(t,0)} = \mathbf{w}^{(t)}$
2: **for** $k = 0, 1 \ldots, \tau - 1$ **do**
3:     Compute stochastic gradient $\nabla f_i(\mathbf{w}_i^{(t,k)}, \xi_i^{(t,k)})$
4:     $\mathbf{w}_i^{(t,k+1)} = \mathbf{w}_i^{(t,k)} - \eta_c\nabla f_i(\mathbf{w}_i^{(t,k)}, \xi_i^{(t,k)})$
5: **end for**
6: Return $(\mathbf{w}^{(t)} - \mathbf{w}_i^{(t,\tau)})/\eta_c\tau$

---

Note that due to the data heterogeneity, randomly sampling $\mathcal{S}^{(t)}$ inherently introduces some variance within our FL system, which we term as the *partial participation error*. We characterize the effect of this partial participation error on the convergence bound of `FedAvg` in the next section.

## 2.1 CONVERGENCE ANALYSIS OF FEDAVG

Before stating our convergence bound, we make the following standard assumptions.

**Assumption 1.** *(Smoothness). Each local objective function is L-Lipshitz smooth, that is, $\|\nabla f_i(\mathbf{x}) - \nabla f_i(\mathbf{y})\| \leq L\|\mathbf{x} - \mathbf{y}\|$, for all $i \in [N]$.*

**Assumption 2.** *(Unbiased gradient and bounded local variance). The stochastic gradient at each client is an unbiased estimator of the local gradient, i.e., $\mathbb{E}_{\xi_i \sim \mathcal{D}_i}[\nabla f_i(\mathbf{w}, \xi_i)] = \nabla f_i(\mathbf{w})$ and its variance is bounded $\mathbb{E}_{\xi_i \sim \mathcal{D}_i}\|\nabla f_i(\mathbf{w}, \xi_i) - \nabla f_i(\mathbf{w})\|^2 \leq \sigma^2$, for all $i \in [N]$.*

**Assumption 3.** *(Bounded global variance). There exists a constant $\sigma_g > 0$ such that the difference between the local gradient at the i-th client and the global gradient is bounded as follows: $\|\nabla f_i(\mathbf{w}) - \nabla f(\mathbf{w})\|^2 \leq \sigma_g^2$, for all $i \in [N]$.*

Following previous work [McMahan et al., 2017, Karimireddy et al., 2019, Wang et al., 2020], we model partial client participation as uniformly sampling a subset of clients *without replacement* from the total pool of clients.

**Theorem 1** (FedAvg Error Decomposition)**.** *Under Assumptions 1, 2, 3, suppose in each round the server randomly selects $M$ out of $N$ clients without replacement to perform $\tau$ steps of local SGD. If the client learning rate $\eta_c$, and the server learning rate $\eta_s$ are chosen such that $\eta_c \leq \frac{1}{8L\tau}$, $\eta_s\eta_c \leq \frac{1}{24\tau L}$, then the iterates $\{\mathbf{w}^{(t)}\}$ generated by `FedAvg` satisfy*

$$\min_{t \in \{0,\ldots,T-1\}} \mathbb{E}\left\|\nabla f(\mathbf{w}^{(t)})\right\|^2$$

$$\leq \mathcal{O}\left(\frac{f(\mathbf{w}^{(0)}) - f^*}{\eta_s\eta_c\tau T}\right) + \underbrace{\mathcal{O}\left(\frac{\eta_s\eta_c L\sigma^2}{M} + \eta_c^2 L^2(\tau-1)\sigma^2\right)}_{\textit{stochastic gradient error}}$$

$$+ \underbrace{\mathcal{O}\left(\frac{\eta_s\eta_c\tau L(N-M)\sigma_g^2}{M(N-1)}\right)}_{\textit{partial participation error}} + \underbrace{\mathcal{O}\left(\eta_c^2 L^2\tau(\tau-1)\sigma_g^2\right)}_{\textit{client drift error}},$$

*where $f^* = \arg\min_{\mathbf{x}} f(\mathbf{x})$.*

**Remark 1.** *Our result shows that the total error floor of `FedAvg` can be decomposed into three distinct sources of error: 1) stochastic gradients; 2) partial client participation; and 3) client drift. Stochastic gradient error arises due to the variance of local gradients (quantified by $\sigma^2$ in Assumption 2) and is unavoidable unless each local objective has a finite sum structure. The cause for both partial participation error and the client drift error lies in data-heterogeneity present among clients (quantified by $\sigma_g$ in Assumption 3). Setting $M = N$ (full participation) gets rid of the error due to partial participation. Similarly, setting $\tau = 1$ (`FedSGD`) eliminates the client drift error.*

Our analysis closely follows [Wang et al., 2020] with the difference that we sample clients without replacement instead of sampling with replacement. A full proof is provided in the supplementary material for completeness.

**Corollary 1.** *Setting $\eta_c = \frac{1}{\sqrt{T}\tau L}$ and $\eta_s = \sqrt{\tau M}$, `FedAvg` converges to a stationary point of the global objective $f(\mathbf{w})$ at a rate given by,*

$$\min_{t \in \{0,\ldots,T-1\}} \mathbb{E}\left\|\nabla f(\mathbf{w}^{(t)})\right\|^2$$

$$\leq \underbrace{\mathcal{O}\left(\frac{1}{\sqrt{M\tau T}}\right)}_{\textit{stochastic gradient error}} + \underbrace{\mathcal{O}\left(\sqrt{\frac{\tau}{MT}}\right)}_{\textit{partial participation error}} + \underbrace{\mathcal{O}\left(\frac{1}{T}\right)}_{\textit{client drift error}}$$

**Remark 2.** *Note that in this case the convergence rate of `FedAvg` is dominated by the error due to partial participation resulting in the leading $\mathcal{O}\left(\sqrt{\frac{\tau}{MT}}\right)$ term whereas client drift error decays at a much faster $\mathcal{O}\left(\frac{1}{T}\right)$ rate. This is primarily due to the fact that client drift error is scaled by $\eta_c^2$ whereas the partial participation error is scaled by $\eta_s\eta_c\tau$ as seen in Theorem 1. In practice, $\eta_c$ is usually set much smaller than $\eta_s$ and hence the total error due to data-heterogeneity is dominated by the variance due to partial client participation rather than client drift.*

Previous works such as [Karimireddy et al., 2019, Li et al., 2020a, Acar et al., 2021] h ave proposed regularizing the local objectives at clients with a global correction term that prevents client models from drifting towards their local minima. In effect, this regularization *artificially* enforces similarity among the modified client objectives such that the effect of data-heterogeneity ($\sigma_g$) is completely eliminated. However, doing so requires clients to *modify* the local

procedures that they run on their devices to incorporate the global correction term. This either requires additional computation at devices (as in [Acar et al., 2021]) or additional communication between client and server (as in [Karimireddy et al., 2019]). Our goal, on the other hand is to just tackle the variance arising from partial client participation in FL. As a result, our proposed algorithm only modifies the *server update procedure* without requiring clients to perform any additional computation or communication. Since partial participation variance dominates the convergence rate of `FedAvg`, eliminating this variance allows us to enjoy the same rates of convergence as `FedDyn` [Acar et al., 2021] and `SCAFFOLD` [Karimireddy et al., 2019]. We discuss our proposed algorithm and its benefits in greater detail in the next section.

# 3 THE FEDVARP ALGORITHM AND ITS CONVERGENCE ANALYSIS

## 3.1 PROPOSED FEDVARP ALGORITHM

`SAGA` [Defazio et al., 2014] was one of the first variance-reduced SGD algorithms that achieved exponential convergence rate for single node strongly convex optimization by maintaining in memory previously computed gradients for each data point. Inspired by the `SAGA` algorithm [Defazio et al., 2014], we propose a novel algorithm `FedVARP` (Algorithm 2) to tackle variance arising due to partial client participation in FL. The main novelty in `FedVARP` lies in applying the variance reduction correction *globally* at the server without adding any additional computation or communication at clients. We elaborate on further details below.

Similar to `FedAvg`, in each round of `FedVARP`, the server selects a random subset $\mathcal{S}^{(t)}$ of clients that perform `LocalSGD` and send back their updates $\Delta_i^{(t)}$ to the server. Recall that in `FedAvg` the global model is updated just using the average of the $\{\Delta_i^{(t)}\}_{i \in \mathcal{S}^{(t)}}$ (see 2). However this adds a large variance to the `FedAvg` update as client data is heterogeneous and the number of selected clients could be much smaller than the total number of clients $N$. The key to reducing this variance is to *approximate* the updates of the clients that do not participate. We propose that the server use the *latest observed update* for each client as the approximation for its current update. Let $\{\mathbf{y}_i^{(t)}\}_{i=1}^N$ represent a state for each client maintained at the server. After every round, we perform the following update (we initialize $\mathbf{y}_i^{(0)} = \mathbf{0}$ for all $i \in [N]$),

$$\mathbf{y}_j^{(t+1)} = \begin{cases} \Delta_j^{(t)} & \text{if } j \in \mathcal{S}^{(t)} \\ \mathbf{y}_j^{(t)} & \text{otherwise} \end{cases}, \text{ for all } j \in [n] \quad (3)$$

This ensures that $\mathbf{y}_i^{(t)}$ maintains the latest observed update from the $i$-th client in round $t$. Note that this implementation requires the server to maintain $\mathcal{O}(Nd)$ memory which can

be expensive in a federated setting. In Section 4 we outline a more practical algorithm `ClusterFedVARP` to reduce the storage requirement.

Given $\{\mathbf{y}_i^{(t)}\}_{i=1}^N$, we can *reuse* the latest observed updates of *all* clients and $\Delta_i^{(t)}$'s of participating clients to compute a *variance reduced* aggregated update,

$$\mathbf{v}^{(t)} = \frac{1}{|\mathcal{S}^{(t)}|} \sum_{i \in \mathcal{S}^{(t)}} \left( \Delta_i^{(t)} - \mathbf{y}_i^{(t)} \right) + \frac{1}{N} \sum_{j=1}^N \mathbf{y}_j^{(t)}, \quad (4)$$

which is used to update the global model as follows,

$$\mathbf{w}^{(t+1)} = \mathbf{w}^{(t)} - \tilde{\eta}_s \mathbf{v}^{(t)}. \quad (5)$$

---

**Algorithm 2** `FedVARP`
___
1: **Input:** initial model $\mathbf{w}^{(0)}$, server learning rate $\eta_s$, client learning rate $\eta_c$, number of local SGD steps $\tau$, $\tilde{\eta}_s = \eta_s \eta_c \tau$, number of rounds $T$, initial states $\mathbf{y}_i^{(0)} = \mathbf{0}$ for all $i \in [n]$, $\mathbf{y}^{(0)} = \mathbf{0}$
2: **for** $t = 0, 1, \dots, T-1$ **do**
3:     Sample $\mathcal{S}^{(t)} \subseteq [N]$ uniformly without replacement
4:     **for** $i \in \mathcal{S}^{(t)}$ **do**
5:         $\Delta_i^{(t)} \leftarrow \texttt{LocalSGD}(i, \mathbf{w}^{(t)}, \tau, \eta_c)$
6:     **end for**
7:     // At Server:
8:     $\mathbf{v}^{(t)} = \mathbf{y}^{(t)} + \frac{1}{|\mathcal{S}^{(t)}|} \sum_{i \in \mathcal{S}^{(t)}} \left( \Delta_i^{(t)} - \mathbf{y}_i^{(t)} \right)$
9:     $\mathbf{w}^{(t+1)} = \mathbf{w}^{(t)} - \tilde{\eta}_s \mathbf{v}^{(t)}$
10:    $\mathbf{y}^{(t+1)} = \mathbf{y}^{(t)} + \frac{1}{N} \sum_{i \in \mathcal{S}^{(t)}} \left( \Delta_i^{(t)} - \mathbf{y}_i^{(t)} \right)$
11:    //State update
12:    **for** $j \in [N]$ **do**
13:        $\mathbf{y}_j^{(t+1)} = \begin{cases} \Delta_j^{(t)} & \text{if } j \in \mathcal{S}^{(t)} \\ \mathbf{y}_j^{(t)} & \text{otherwise} \end{cases}$
14:    **end for**
15: **end for**

---

Note that `FedVARP` gives higher weight to current client updates as compared to previous client updates which allows it to enjoy the additional *unbiased* property,

$$\mathbb{E}_{\mathcal{S}^{(t)}} \left[ \mathbf{v}^{(t)} \right] = \mathbb{E}_{\mathcal{S}^{(t)}} \left[ \frac{1}{|\mathcal{S}^{(t)}|} \sum_{i \in \mathcal{S}^{(t)}} \Delta_i^{(t)} \right]. \quad (6)$$

This implies that in expectation `FedVARP` performs the same update as `FedAvg`. This simplifies our analysis considerably and allows us to set $\mathbf{y}_i^{(0)} = \mathbf{0}$ without any complications in theory or practice. We further highlight the importance of server-based `SAGA` in comparison to related work.

**Comparison with MIFA.** Closely related to this work, [Gu et al., 2021] proposed the `MIFA` algorithm to deal with arbitrary device unavailability in FL. `MIFA` also maintains in memory the latest observed updates for each client and instead applies a `SAG`-like [Schmidt et al., 2017] aggregation

of these updates. Unlike `FedVARP`, `MIFA` assigns equal weights to both the current and previous updates, making it a biased scheme. This complicates their analysis significantly, which requires additional assumptions such as almost surely bounded gradient noise and Hessian Lipschitzness. Furthermore, due to this bias, `MIFA` requires all the clients to participate in the first round, which is unrealistic in many FL settings. We compare the performance of `FedVARP` with `MIFA` in our experiments (see Section 5) and show that `FedVARP` consistently outperforms `MIFA`.

**Comparison with SCAFFOLD.** `SCAFFOLD` [Karimireddy et al., 2019] is one of the first works to identify the client drift error and it proposes the use of control variates to correct it. This requires clients to apply a `SAGA`-like variance reduction correction at *every local* step. This leads to a 2x rise in communication as the clients now need to communicate both the global model as well as the global correction vector to the server. In `FedVARP`, clients perform `LocalSGD` and are *agnostic* to any aspect of how the variance reduction is applied at the server. This saves the cost of communicating the update to the global correction vector while maintaining the same rate of convergence as `SCAFFOLD`.

Hence, we see that server-based SAGA variance reduction is especially suited for the federated setting. It avoids extra computation or communication at the clients (as in `SCAFFOLD`) or unrealistic client participation scenarios (as in `MIFA`).

### 3.2 CONVERGENCE ANALYSIS OF FEDVARP

**Theorem 2** (Convergence of `FedVARP`). *Suppose the functions $\{f_i\}$ satisfy Assumptions 1, 2, 3. In each round of `FedVARP`, the server randomly selects $|\mathcal{S}^{(t)}| = M$ (out of $N$) clients, for all $t$, without replacement, to perform $\tau$ steps of local SGD. If the server and client learning rates, $\eta_s, \eta_c$ respectively, are chosen such that $\eta_s \eta_c \leq \min\left\{\frac{M^{3/2}}{8L\tau N}, \frac{5M}{48\tau L}, \frac{1}{4L\tau}\right\}$ and $\eta_c \leq \frac{1}{10L\tau}$, then the iterates $\{\mathbf{w}^{(t)}\}$ generated by `FedVARP` satisfy*

$$\min_{t\in\{0,\ldots,T-1\}}\mathbb{E}\left\|\nabla f(\mathbf{w}^{(t)})\right\|^2 \leq \mathcal{O}\left(\frac{f(\mathbf{w}^{(0)}) - f^*}{\eta_s \eta_c \tau T}\right)$$
$$+ \underbrace{\mathcal{O}\left(\frac{\eta_s \eta_c L\sigma^2}{M} + \eta_c^2 L^2(\tau-1)\sigma^2\right)}_{\text{stochastic gradient Error}} + \underbrace{\mathcal{O}\left(\eta_c^2 L^2 \tau(\tau-1)\sigma_g^2\right)}_{\text{client drift error}},$$

*where $f^* = \arg\min_{\mathbf{x}} f(\mathbf{x})$.*

We defer the proof and the exact convergence rate of `FedVARP` to our supplementary material. We observe that `FedVARP` successfully eliminates the partial participation error, while retaining the stochastic sampling error and client drift error. This is to be expected as we do not modify the `LocalSGD` procedure at the clients to control these errors.

**Reduction to SAGA.** Note that in the case when $\sigma = 0$, $\tau = 1$ and $M = 1$ our algorithm reduces exactly to the `SAGA` algorithm [Defazio et al., 2014]. Setting $\eta_c = \frac{1}{8LN}$ and $\eta_s = 1$ we get a rate of $\mathcal{O}\left(\frac{N}{T}\right)$ for non-convex loss functions. Our rate is slightly worse than the rate of $\mathcal{O}\left(\frac{N^{2/3}}{T}\right)$ obtained in [Reddi et al., 2016] because we use the *same* sample $i^{(t)}$ to update both $\mathbf{w}^{(t)}$ and $\mathbf{y}_{i^{(t)}}$. [Reddi et al., 2016] instead draw *two independent* samples $i^{(t)}$ and $j^{(t)}$, where $i^{(t)}$ is used to update the model $\mathbf{w}^{(t)}$ and $j^{(t)}$ is used to update $\mathbf{y}_{j^{(t)}}$. For a fixed $\mathbf{w}^{(t)}$, this effectively ensures independence between $\mathbf{w}^{(t+1)}$ and $\{\mathbf{y}_j^{(t+1)}\}_{j=1}^N$ which we believe leads to the theoretical improvement in their convergence rates.

## 4 CLUSTER FEDVARP, AND ITS CONVERGENCE ANALYSIS

While `FedVARP` successfully eliminates partial client participation variance, it does so at the expense of maintaining a $\mathcal{O}(Nd)$ memory of latest client updates at the server. This storage cost can quickly become prohibitive since both $N$ and $d$ can be large in federated settings [Kairouz et al., 2019, Reddi et al., 2021]. To remedy this, we propose `ClusterFedVARP`, a novel server-based aggregation strategy to reduce partial client participation variance while being storage-efficient.

`ClusterFedVARP` is based on the simple observation that we can reduce storage cost by partitioning our set of $N$ clients into $K$ disjoint clusters and maintaining a *single* state for all the clients in the same cluster. In other words, instead of maintaining $N$ states for $N$ clients, we maintain just $K$ cluster states with clients in the same cluster *sharing* the same state. Assuming that there exists such a clustering of clients, our algorithm proceeds as follows. Let $c_i \in [K]$ be the cluster identity of the $i$-th client. We initialize all cluster states to zero, that is, $\mathbf{y}_k^{(0)} = \mathbf{0}$ for all $k \in [K]$. Different from `FedVARP`, we now use the cluster states of clients to compute $\mathbf{v}^{(t)}$, i.e.,

$$\mathbf{v}^{(t)} = \frac{1}{|\mathcal{S}^{(t)}|}\sum_{i\in\mathcal{S}^{(t)}}\left(\Delta_i^{(t)} - \mathbf{y}_{c_i}^{(t)}\right) + \frac{1}{N}\sum_{j=1}^N \mathbf{y}_{c_j}^{(t)}. \quad (7)$$

We observe that $\mathbf{v}^{(t)}$ still enjoys the unbiased property outlined in 6 since,

$$\mathbb{E}_{\mathcal{S}^{(t)}}\left[\frac{1}{|\mathcal{S}^{(t)}|}\sum_{i\in\mathcal{S}^{(t)}}\mathbf{y}_{c_i}^{(t)}\right] = \frac{1}{N}\sum_{j=1}^N \mathbf{y}_{c_j}^{(t)} \quad (8)$$

The major algorithmic difference lies in how we update the cluster states,

$$\mathbf{y}_k^{(t+1)} = \begin{cases} \frac{\sum_{i\in\mathcal{S}^{(t)}\cap\mathcal{C}_k}\Delta_i^{(t)}}{|\mathcal{S}^{(t)}\cap\mathcal{C}_k|} & \text{if } |\mathcal{S}^{(t)}\cap\mathcal{C}_k| \neq 0, \\ \mathbf{y}_k^{(t)} & \text{otherwise,} \end{cases} \quad (9)$$

for all $k \in [K]$. For $k$-th cluster $\mathcal{C}_k$, the cluster state is the *average* update of the participating clients that belong to cluster $k$, i.e., $\mathcal{S}^{(t)} \cap \mathcal{C}_k$. If this set is empty the cluster state remains unchanged.

---

**Algorithm 3** `ClusterFedVARP`

---

1: **Input:** initial model $\mathbf{w}^{(0)}$, server learning rate $\eta_s$, client learning rate $\eta$, local SGD steps $\tau$, $\tilde{\eta}_s = \eta_s \eta_c \tau$, number of rounds $T$, number of clusters $K$, initial cluster states $\mathbf{y}_k^{(0)} = \mathbf{0}$ for all $k \in [K]$, cluster identities $c_i \in [K]$ for all $i \in [N]$, cluster sets $\mathcal{C}_k = \{i : c_i = k\}$ for all $k \in [K]$
2: **for** $t = 1, 2, \ldots, T$ **do**
3:    Sample $\mathcal{S}^{(t)} \subseteq [N]$ uniformly without replacement
4:    **for** $i \in \mathcal{S}^{(t)}$ **do**
5:        $\Delta_i^{(t)} \leftarrow \text{LocalSGD}(i, \mathbf{w}^{(t)}, \tau, \eta)$
6:    **end for**
7:    // At Server:
8:    $\mathbf{v}^{(t)} = \frac{1}{|\mathcal{S}^{(t)}|} \sum_{i \in \mathcal{S}^{(t)}} \left( \Delta_i^{(t)} - \mathbf{y}_{c_i}^{(t)} \right) + \frac{1}{N} \sum_{j=1}^{N} \mathbf{y}_{c_j}^{(t)}$
9:    $\mathbf{w}^{(t+1)} = \mathbf{w}^{(t)} - \tilde{\eta}_s \mathbf{v}^{(t)}$
10:    //State update
11:    **for** $k \in [K]$ **do**
12:    $\mathbf{y}_k^{(t+1)} = \begin{cases} \dfrac{\sum_{i \in \mathcal{S}^{(t)} \cap \mathcal{C}_k} \Delta_i^{(t)}}{|\mathcal{S}^{(t)} \cap \mathcal{C}_k|} & \text{if } |\mathcal{S}^{(t)} \cap \mathcal{C}_k| \neq 0 \\ \mathbf{y}_k^{(t)} & \text{otherwise} \end{cases}$
13:    **end for**
14: **end for**

---

Note that the dissimilarity in client data across clusters is already bounded in Assumption 3. Our motivation behind using a clustering approach is to utilize a tighter bound on the data dissimilarity *within* a cluster. We quantify this precisely via the following assumption.

**Assumption 4.** *(Bounded cluster variance). Let $K$ be the total number of clusters and $\mathcal{C}_k$ be the set of clients belonging to the $k$-th cluster . There exists a constant $\sigma_K \geq 0$ such that the difference between the average gradient of clients in the $k$-th cluster and the local gradient of the $i$-th client in the $k$-th cluster is bounded as follows:* $\left\| \nabla f_i(\mathbf{w}) - \frac{1}{|\mathcal{C}_k|} \sum_{j \in \mathcal{C}_k} \nabla f_j(\mathbf{w}) \right\|^2 \leq \sigma_K^2$, *for all $k \in [K]$, for all $i \in \mathcal{C}_k$.*

We see that $\sigma_K^2$ acts a measure of the efficacy of our clustering with the goal being to achieve $\sigma_K^2 \ll \sigma_g^2$. In practice, there often exists metadata about clients that can be used to naturally partition clients into well-structured clusters. For instance, when training a next-word prediction model [Hard et al., 2018], clients could be grouped by geographical location depending on the local dialect. Another example is training recommender systems for social media platforms [Jalalirad et al., 2019] where we expect connected users to have similar interests.

Intuitively, we expect that for $K < N$ we will suffer an error of $\mathcal{O}\left(\sigma_K^2\right)$ when trying to approximate a client's update by

its cluster state. This intuition is captured precisely in our convergence result for `ClusterFedVARP` as stated below.

**Theorem 3** (Convergence of `ClusterFedVARP`). *Suppose the functions $\{f_i\}$ satisfy Assumptions 1, 2, 3, 4. Further, suppose all the clients are partitioned into $K$ clusters, each with $r$ clients, such that $N = rK$. In each round of `ClusterFedVARP`, the server randomly selects $|\mathcal{S}^{(t)}| = M$ (out of $N$) clients, for all $t$, without replacement, to perform $\tau$ steps of local SGD. Further, the client learning rate $\eta_c$, and the server learning rate $\eta_s$ are chosen such that $\eta_c \leq \frac{1}{10L\tau}$, $\eta_s \eta_c \leq \min\left\{ \frac{\sqrt{M}(1-p)}{8L\tau}, \frac{M}{16\tau L}, \frac{1}{4L\tau} \right\}$, where $p = \frac{\binom{N-r}{M}}{\binom{N}{M}}$. Then, the iterates $\{\mathbf{w}^{(t)}\}_t$ generated by `ClusterFedVARP` satisfy*

$$\min_{t \in \{0, \ldots, T-1\}} \mathbb{E} \left\| \nabla f(\mathbf{w}^{(t)}) \right\|^2$$

$$\leq \mathcal{O}\left( \frac{f(\mathbf{w}^{(0)}) - f^*}{\eta_s \eta_c \tau T} \right) + \underbrace{\mathcal{O}\left( \frac{\eta_s \eta_c L \sigma^2}{M} + \eta_c^2 L^2 (\tau-1)\sigma^2 \right)}_{\text{stochastic sampling error}}$$

$$\underbrace{\mathcal{O}\left( \frac{\eta_s \eta_c L \tau (N-M) \sigma_K^2}{M(N-1)} \right)}_{\text{cluster heterogeneity error}} + \underbrace{\mathcal{O}\left( \eta_c^2 L^2 \tau (\tau-1) \sigma_g^2 \right)}_{\text{client drift error}}$$

We defer the proof and exact convergence rate to our supplementary material. For $K = N$ (one client per cluster) we recover the convergence rate of `FedVARP`($\sigma_{K=N}^2 = 0$). On the other hand, for $K = 1$ we get back the `FedAvg` algorithm since all clients share the same state and there is no variance-reduction ($\sigma_{K=1}^2 = \sigma_g^2$). Thus, we see a natural trade-off between storage and variance-reduction as we vary the number of cluster states $K$. In practice, `ClusterFedVARP` gives server the flexibility to set $K$ based on its storage constraints.

We see that `ClusterFedVARP` also allows an interesting trade-off between the server learning rate and cluster approximation error as we vary $K$. Our analysis shows the bound on the server learning rate comes from trying to control the "staleness" of a client's state, which measures the frequency with which a client's state is updated. In `FedVARP`, a client's state is updated only when the client participates, which happens with probability $\frac{M}{N}$. In `ClusterFedVARP` a client's state is updated as long as *any* client from the same cluster participates, which dramatically reduces staleness. However this comes at the cost of the additional cluster heterogeneity error implying a trade-off between convergence speed and error floor.

## 5 EXPERIMENTS

### 5.1 EXPERIMENTAL SETUP

To support our theoretical findings we evaluate our proposed algorithms on the following FL tasks: i) image classification on CIFAR-10 [Krizhevsky et al., 2009] with LeNet-5

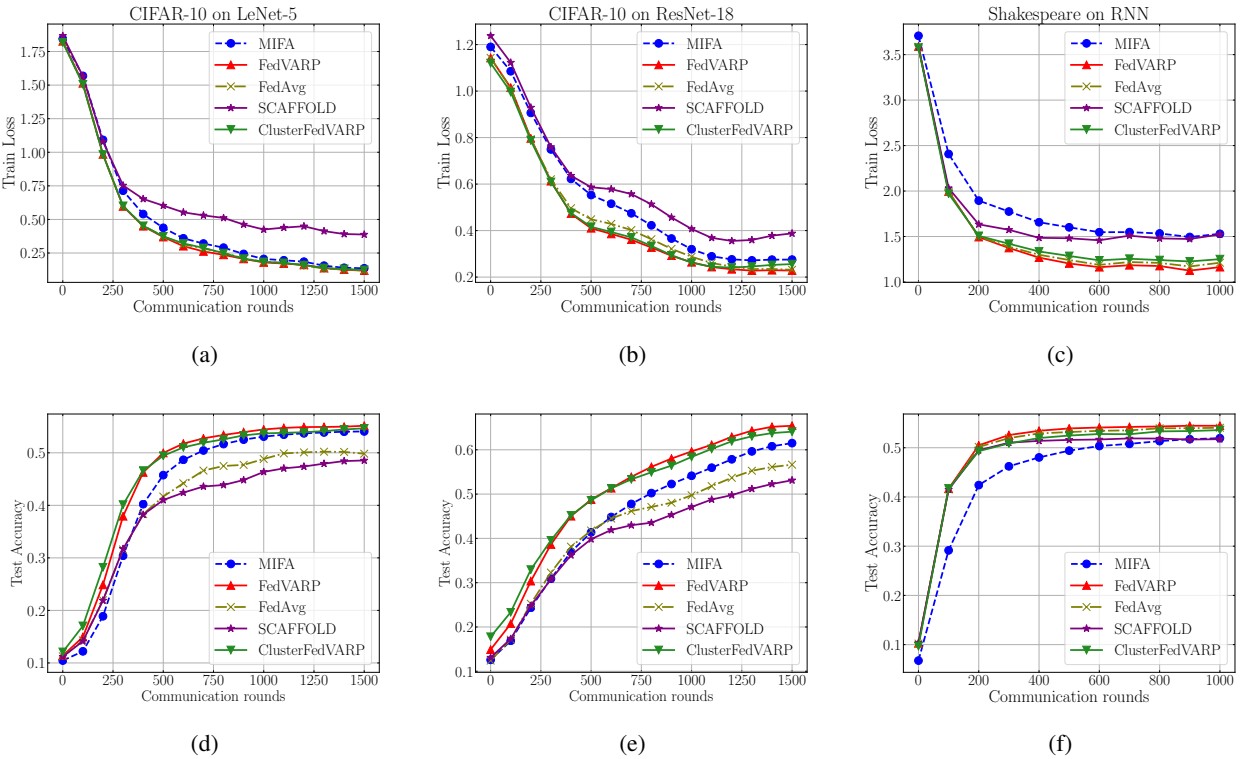

Figure 1: Experimental Results showing Training Loss and Test Accuracy for: CIFAR-10 on LeNet-5 (a,d), CIFAR-10 on ResNet-18 (b,e), Shakespeare on RNN (c,f). For `ClusterFedVARP` we keep $K = 55$ for CIFAR-10 experiments (4.5x storage reduction) and $K = 36$ for Shakespeare experiments (30x storage reduction). `FedVARP` outperforms baselines in all cases while `ClusterFedVARP` outperforms baselines in most cases. We see greater empirical benefits for CIFAR-10 experiments due to the higher data-heterogeneity across clients.

[LeCun et al., 2015], ii) image classification on CIFAR-10 with ResNet-18 [He et al., 2016], and iii) next character prediction on Shakespeare [Caldas et al., 2018] with a RNN model. In all setups, we compare the performance of our algorithms with `FedAvg`, `MIFA` [Gu et al., 2021] and `SCAFFOLD` [Karimireddy et al., 2019] (see Section 3.1 for discussion of the algorithms). We briefly describe the datasets and the natural clustering of clients that we utilize in these datasets.

**CIFAR-10.** The CIFAR-10 dataset is a natural image dataset consisting of 60000 32x32 colour images, with each image assigned to one of 10 classes (6000 images per class). We create a federated non-iid split of the CIFAR-10 dataset among 250 clients using a similar procedure as [McMahan et al., 2017]. The data is first sorted by labels and divided into 500 shards with each shard corresponding to data of a particular label. Clients are randomly assigned 2 such shards which implies each client has a data distribution corresponding to either 1 or 2 classes. For `ClusterFedVARP`, we group clients having the same data distribution in the same cluster giving us 55 unique clusters.

**Shakespeare.** Shakespeare is a language modelling task

where each client is a role from one of the plays in *The Collective Works of William Shakespeare* [Shakespeare, 2014]. We pick clients that have lines corresponding to at least 120 characters which leaves us with 1089 unique clients. The task is to predict the next character given an input sequence of 20 characters from a client's text. For `ClusterFedVARP`, we group clients belonging to the same play in the same cluster giving us a total of 36 clusters.

**Experimental Details.** To simulate partial client participation we uniformly sample $M = 5$ clients without replacement in every round for all algorithms. This gives us a participation rate of 2% for CIFAR-10 experiments and $< 1\%$ for Shakespeare as seen in practice for typical FL settings [Kairouz et al., 2019]. We allow clients to perform 5 local epochs before sending their updates. We use a batch size of 64 in all experiments. We fix the server learning rate $\eta_s$ to 1 and tune the client learning rate $\eta_c$ over the grid $\{10^{-1}, 10^{-1.5}, 10^{-2}, 10^{-2.5}, 10^{-3}\}$ for all algorithms. For ResNet-18 we replace the batch normalization layers by group normalization [Hsieh et al., 2020]. Our Shakespeare RNN was a single layer Gated Recurrent Unit (GRU) with 128 hidden parameters and embedding dimension of 8.

## 5.2 COMPARISON WITH BASELINES

Our experiments clearly demonstrate that our proposed algorithms consistently outperform other baselines without requiring additional communication or computation at clients. `ClusterFedVARP` closely matches the performance of `FedVARP` in all experiments thereby highlighting the practical gains of clustering-based storage reduction. For instance, to achieve 50% test accuracy on CIFAR-10 classification with LeNet-5 our algorithms take less than 536 rounds while `FedAvg` takes 1158 rounds giving us up to 2.1x speedup. The benefits are especially pronounced for CIFAR-10 as the artificial data partitioning leads to greater heterogeneity across clients thereby accentuating the effect of partial participation.

Our algorithms also outperform competing variance-reduction methods `MIFA` and `SCAFFOLD` in all experiments. The performance of `MIFA` is severely affected by its bias in the initial rounds of training since we do not assume that all clients participate in the first round of training. This again highlights the practical usefulness of the unbiased variance-reduction applied in `FedVARP` and `ClusterFedVARP`. While theoretically appealing we find that modifying the `LocalSGD` procedure using `SCAFFOLD` to mitigate client drift actually hurts performance in practical FL settings. Our findings are consistent with [Reddi et al., 2021] and make the case for reducing client drift using carefully tuned local learning rates while focusing on server-based optimization techniques to reduce variance.

## 6 RELATED WORK

**Convergence Analysis of `FedAvg`:** The original `FedAvg` [McMahan et al., 2017] work inspired a rich line of work trying to analyze `FedAvg` in various settings [Khaled et al., 2020, Yu et al., 2019, Li et al., 2020b]. The convergence results closest to our setting are found in [Wang et al., 2020, Karimireddy et al., 2019, Yang et al., 2021] that analyze `FedAvg` in the presence of non-iid data as well as partial client participation for non-convex objectives. We refer readers to [Kairouz et al., 2019, Wang et al., 2021] for a comprehensive review of convergence results in FL.

**Variance Reduction.** Since the inception of SAG [Schmidt et al., 2017] and SAGA [Defazio et al., 2014], several variance-reduction methods for centralized stochastic problems have been proposed that do not require additional storage. We divide these works into two broad categories and discuss applying them in a federated context to reduce partial client participation.

1) **SVRG-style Variance Reduction.** SVRG [Johnson and Zhang, 2013] and related methods like SCSG [Lei et al., 2017] SARAH [Nguyen et al., 2017], and SPIDER [Fang et al., 2018] trade-off storage with computation and need to compute the full (or a large-minibatch) gradient at regular intervals. While these methods achieve theoretically better rates than SAGA, applying them in a federated context would require *all* clients to participate in some rounds of training which we believe is unrealistic.

2) **Momentum-based Variance Reduction.** A recent line of work explores the connection between SGD with momentum and variance-reduction and proposes new algorithms STORM [Cutkosky and Orabona, 2019] and HybridSARAH [Tran-Dinh et al., 2019], that do not require full-batch gradient computation at any iteration. This has inspired federated counterparts [Das et al., 2020], [Khanduri et al., 2021], [Li et al., 2021]. [Das et al., 2020] and [Li et al., 2021] propose to use such approaches to reduce client participation variance. However there are two drawbacks. The central server needs to communicate two sets of global models $\mathbf{w}^{(t)}$ and $\mathbf{w}^{(t-1)}$ to the participating clients, doubling server to client communication. Secondly, participating clients need to run local SGD for both sets of global models, thereby doubling computation. Again while theoretically attractive we believe such approaches are not suitable for practical FL settings.

**Clustered Federated Learning and Variance Reduction.** The idea of utilizing cluster structure among clients has given rise to the paradigm of *clustered federated learning* [Ghosh et al., 2020], [Sattler et al., 2020], where *separate* global models are learned for each cluster. On the other hand, we propose to learn a *single* global model and use the cluster structure for reducing the variance arising due to partial client participation. A similar idea of sharing gradient information while reducing variance has been explored in $\mathcal{N}$-SAGA [Hofmann et al., 2015] but their focus is on a single node centralized setting and the analysis is restricted to strongly convex functions. An interesting direction for future work is to linearly combine a client's previous state with its cluster state to reduce staleness as done in [Allen-Zhu et al., 2016].

## 7 CONCLUSION

We consider the problem of eliminating variance arising due to partial client participation in large-scale FL systems. We first show that partial participation variance dominates the convergence rate of `FedAvg` for smooth non-convex loss functions. We propose `FedVARP`, a novel aggregation strategy applied at the server to completely eliminate this variance without requiring any additional computation or communication at the clients. Next we propose a more practical clustering-based strategy `ClusterFedVARP` that reduces variance while being storage-efficient. Our theoretical findings are comprehensively supported by our experimental results which show that our proposed algorithms consistently outperform existing baselines.

**Acknowledgements**

This research was generously supported in part by the NSF Award (CNS-2112471), the NSF CAREER Award (CCF-2045694), and the David H. Barakat and LaVerne Owen-Barakat College of Engineering Dean's Fellowship at Carnegie Mellon University.

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
