# OpenReview forum: "FedVARP: Tackling the Variance Due to Partial Client Participation in Federated Learning"
_auai.org/UAI/2022/Conference — UAI 2022 Poster_

### Official Review · Reviewer_wiBd · 2022-03-30

**Q2(1) Originality/Novelty:** 3
**Q2(2) Significance/Impact:** 3
**Q2(3) Correctness/Technical Quality:** 3
**Q2(6) Clarity Of Writing:** 4
**Q6 Overall Score:** 6
**Q8 Confidence In Your Score:** 4

**Q1 Summary And Contributions:**

This paper studies the problem of federated learning. Given two error sources in FL,
clients drift due to multiple local updates and partial participation caused by sampling,
the paper aims to reduce the error of the first type. To this end, an algorithm FEDVARP is developed, which keeps the local
sgd and communication scheme intact but changes the aggregation strategy at the server-side.
Theorem 2 shows that FEDVARP removes the error of drift, but requires a memory space of size N by d.


**Q2 Assessment Of The Paper:**

More detailed information regarding each of these aspects is given below:

**Q2(4) Quality Of Experiments (Optional):**

3: Good: The experimental evaluation is adequate, and the results convincingly support the main claims.

**Q2(5) Reproducibility:**

3: Good: Key resources (e.g., proofs, code, data) are available and key details (e.g., proofs, experimental setup) are sufficiently well-described for competent researchers to confidently reproduce the main results.

**Q3 Main Strengths:**

I like the idea of FEDVARP. The main idea is to let the clients do some usual computations and keep the communication scheme simple, but make the server to remove the errors.

Theorem 1 nicely decomposes the errors in different groups and serves as a nice building block to the rest of the paper.

FEDVARP although quite similar to SAGA, is shown that can successfully remove the error of client drifts. I do have some concerns that how large this error can be compared to the other error terms.

The clustering idea is also applied as a normal way to reduce the memory requirements.

**Q4 Main Weakness:**

The literature review can be greatly improved. A comparison with SCAFFOLD is given but it should contain more details on the differences and should provide a comparison on the rates. FL works based on importance sampling are also missing.

Since the first algorithm is similar to SAGA and the second one uses an expected technique of clustering, I wonder how novel are the technical analysis here? What is the main difficulty in the analysis?

The experiments show the superiority of almost all schemes over the SCAFFOLD. This is surprising. An explanation should be added to justify this observation.


**Q5 Detailed Comments To The Authors:**

Please check the comments above

**Q7 Justification For Your Score:**

check the weaknesses

**Q9 Complying With Reviewing Instructions:**

1: Yes.

---

### Official Review · Reviewer_4Wez · 2022-04-11

**Q2(1) Originality/Novelty:** 3
**Q2(2) Significance/Impact:** 3
**Q2(3) Correctness/Technical Quality:** 3
**Q2(6) Clarity Of Writing:** 3
**Q6 Overall Score:** 7
**Q8 Confidence In Your Score:** 3

**Q1 Summary And Contributions:**

The paper considers the problems in data-heterogeneous federated learning systems and proposes FedVARP, a variance reduction algorithm, to eliminate errors in partial client partition.  ClusterFedVARP is further proposed to alleviate the memory cost.  Theoretical analysis and experiments are provided to show the effectiveness of the proposal.

**Q2 Assessment Of The Paper:**

More detailed information regarding each of these aspects is given below:

**Q2(4) Quality Of Experiments (Optional):**

3: Good: The experimental evaluation is adequate, and the results convincingly support the main claims.

**Q2(5) Reproducibility:**

3: Good: Key resources (e.g., proofs, code, data) are available and key details (e.g., proofs, experimental setup) are sufficiently well-described for competent researchers to confidently reproduce the main results.

**Q3 Main Strengths:**

1. The paper considers an interesting problem to tackle problems in data-heterogeneous federated learning systems.
2. A suitable proposal, FedVARP and ClusterFedVARP, are presented with theoretical proof and empirical evaluation.

**Q4 Main Weakness:**

1. The work seems an extension of Variance Reduction for Federated Learning.  Most of the derivation follows the way of Variance Reduction.
2. Some experimental results can be enriched to understand the performance scope of the proposed algorithms.

**Q5 Detailed Comments To The Authors:**

Overall, the paper has clearly presented the VAriance Reduction algorithm for Federated Learning.  Both theoretical justification and empirical evaluation are provided.  Some more issues about the experiments can be added to enrich the performance scope of the proposed algorithms.  For example,

1. An ablation study on ClusterFedVARP, e.g., the number of clusters, is encouraged.
2. It is interesting to show the standard deviation of the compared methods in Fig. 1.  So that we can know the robustness of the corresponding methods.

**Q7 Justification For Your Score:**

Overall, the paper is clearly presented with both theoretical justification and empirical evaluation, though the empirical results can be enriched.



**Q9 Complying With Reviewing Instructions:**

1: Yes.

---

### Official Review · Reviewer_G1cK · 2022-04-11

**Q2(1) Originality/Novelty:** 3
**Q2(2) Significance/Impact:** 2
**Q2(3) Correctness/Technical Quality:** 3
**Q2(6) Clarity Of Writing:** 4
**Q6 Overall Score:** 7
**Q8 Confidence In Your Score:** 4

**Q1 Summary And Contributions:**

The paper introduces FedVARP and ClusterFedVARP two variations on FedAvg that deal with the problem of reducing the error due to the fact that not all clients participate in each round. The authors show that, under given (mild) assumptions, the error is dominated indeed by this kind of error and propose FedVARP and ClusterFedVarp to solve this problem. The algorithms are supported by solid theory and empirical evidence.

**Q2 Assessment Of The Paper:**

More detailed information regarding each of these aspects is given below:

**Q2(4) Quality Of Experiments (Optional):**

3: Good: The experimental evaluation is adequate, and the results convincingly support the main claims.

**Q2(5) Reproducibility:**

2: Fair: Key resources (e.g., proofs, code, data) are unavailable but key details (e.g., proof sketches, experimental setup) are sufficiently well-described for an expert to confidently reproduce the main results.

**Q3 Main Strengths:**

The paper is very well written and proposes an original idea. While the code to reproduce the experiments is not provided, the proofs are provided in full in the appendix.

**Q4 Main Weakness:**

As mentioned the paper is very well written. If pressed I would have liked to see some discussion about how the proposed work would fit in a cross-silos setting, where many of the assumptions in the paper are either invalid or valid to a lesser degree.

**Q5 Detailed Comments To The Authors:**

I found the paper enjoyable to read. As mentioned above, the only nitpicking detail that I would point out is that the paper implicitly assumes that all FL is cross-device. Cross-silos settings are, IMHO, not only to be disregarded but of increasing relevance due to the new legislation that is being created and due to the adoption of these technologies in the industry and public services (e.g., hospitals).

I believe that, contrary to the provided analysis, in a cross-silos setting the impact of client drift might be of major impact. In this case in fact, the number of clients is very small, they almost always participate and the differences in the dataset can be lareger w.r.t. the cross-device settings. This changes the assumptions made in Corollary 1 that make us conclude that the client drift error is less important.

However, as I mentioned this is a nitpicking detail, the authors never explicitly claim that their analysis would work in a cross-silos setting and implicitly assume a cross-device setting. Everything can be solved for good by just mentioning this assumption in the introduction of the paper.

**Q7 Justification For Your Score:**

This is a good paper, well written, and theoretically well supported. It is hard to argue for a better score since it would require impacts on more than one area of AI and I don't think this is likely. One could argue for a 6, but I loved the presentation and I think the authors deserve something more than a weak accept.

**Q9 Complying With Reviewing Instructions:**

1: Yes.

---

### Decision · Program_Chairs · 2022-05-15

**Decision:**

Accept (Poster)

**Comment:**

Meta Review: The paper proposes a variance reduction method to address partial client participation in federated learning. The reviewers thought the contribution was interesting and addressed an important aspect of federated learning.